# Self-Assembled Metal–Organic Biohybrids (MOBs) Using Copper and Silver for Cell Studies

**DOI:** 10.3390/nano9091282

**Published:** 2019-09-08

**Authors:** Neha Karekar, Anik Karan, Elnaz Khezerlou, Neela Prajapati, Chelsea D. Pernici, Teresa A. Murray, Mark A. DeCoster

**Affiliations:** 1Cellular Neuroscience Laboratory, Molecular Science and Nanotechnology, Applied and Natural Sciences, Louisiana Tech University, Ruston, LA 71270, USA; 2Cellular Neuroscience Laboratory, Biomedical Engineering, College of Engineering and Sciences, Louisiana Tech University, Ruston, LA 71270, USA (A.K.) (E.K.) (N.P.) (T.A.M.); 3College of Pharmacy, University of Utah, Salt Lake City, UT 84112, USA; 4Cellular Neuroscience Laboratory, Institute for Micromanufacturing, College of Engineering and Sciences, Louisiana Tech University, Ruston, LA 71270, USA

**Keywords:** self-assembly, amino acid, copper-containing high-aspect ratio structures (CuHARS), silver nanoparticles, anti-cancer, cystine-capped nanoparticles, functionalization

## Abstract

The novel synthesis of metal-containing biohybrids using self-assembly methods at physiological temperatures (37 °C) was compared for copper and silver using the amino acid dimer cystine. Once assembled, the copper containing biohybrid is a stable, high-aspect ratio structure, which we call CuHARS. Using the same synthesis conditions, but replacing copper with silver, we have synthesized cystine-capped silver nanoparticles (AgCysNPs), which are shown here to form stable colloid solutions in contrast to the CuHARS, which settle out from a 1 mg/mL solution in 90 min. Both the copper and silver biohybrids, as synthesized, demonstrate very low agglomeration which we have applied for the purpose of applications with cell culture methods, namely, for testing as anti-cancer compounds. AgCysNPs (1000 ng/mL) demonstrated significant toxicity (only 6.8% viability) to glioma and neuroblastoma cells in vitro, with concentrations as low as 20 ng/mL causing some toxicity. In contrast, CuHARS required at least 5 μg/mL. For comparative purposes, silver sulfate at 100 ng/mL decreased viability by 52% and copper sulfate at 100 ng/mL only by 19.5% on glioma cells. Using these methods, the novel materials were tested here as metal–organic biohybrids (MOBs), and it is anticipated that the functionalization and dynamics of MOBs may result in building a foundation of new materials for cellular applications, including cell engineering of both normal and diseased cells and tissue constructs.

## 1. Introduction

Bottom-up self-assembly of nanomaterials in general has the advantage of scalability for a given product, with the concept being that once synthesis conditions have been defined, for the case of liquid systems, the volume of the synthesis production may be increased to give a greater yield. We have utilized this concept to translate our discovery of copper-containing high-aspect ratio structures (CuHARS) into methods for characterization and concentration of the materials [1,2,3]. Using the same biological conditions carried out to generate CuHARS, we show here that silver nanoparticles capped with the amino acid dimer cystine may also be produced, consistent with the growing literature providing evidence for many silver nanoparticle capping/reducing agents including ascorbic acid, polysaccharides, and lactic acid bacteria [4,5,6]. As an immediate application for both synthesized materials, we used CuHARS and AgCysNPs as anti-cancer agents against cancer cells in vitro. A major materials advantage of these metal-containing biohybrids (MOBs) was their low agglomeration properties. This was compared to and in contrast to copper oxide nanoparticles (CuNPs), which have previously been shown to agglomerate quickly, and to be very reactive under cell culture conditions [1], and to copper free metal nanoparticles (CuMNPs), which have not been oxidized. We utilized copper sulfate and silver sulfate as the metal sources for our synthesis, and therefore also tested the potential toxicity of these compounds on glioma and neuroblastoma cells. Some limited work has been done previously on evaluating these metal salts in cell culture models [7,8], and in the case of copper, endogenous chelating agents such as albumin [9] and ceruloplasmin [10] may moderate the potential toxicity. Finally, due to the particulate nature, but varying shape and reactivity characteristics of CuNPs, CuHARS, and AgCysNPs, we compared all three materials for establishing the well-known coffee ring effect [11]. This effect was used here to establish MOBs nanofilms on cell culture plates for testing cell-materials interactions using cancer cells. By immobilizing MOBs onto cell culture surfaces in the form of nanofilms, a starting point was established for evaluating the diffusion and breakdown of materials for cell delivery as well as providing an environment for testing cellular outcomes over time. This novel approach developed here of layering and drying materials first onto the surface to be tested for toxicity and then introducing cells, was compared to the traditional method of plating cells first, followed by addition of nanomaterials on top of them [1,12,13]. 

## 2. Materials and Methods 

### 2.1. Cell Culture

(1) CRL-2303 glioma cells were maintained, as suggested by the vendor (ATCC, Manassas, VA, USA), in complete growth medium as previously described [14]. 

(2) SH-EP1a neuroblastoma cells were maintained, as suggested by the vendor (ATCC), in complete growth medium consisting of a 1:1 ratio of Eagle’s Minimal Essential Medium (VWR Life Science, Sanborn, NY, USA) and F12 Medium (Sigma Life Science, St. Louis, MO, USA), with 10% added fetal bovine serum (VWR Seradigm Life Science, Radnor, PA, USA). 

### 2.2. Synthesis of MOBs

CuHARS: CuHARS were synthesized and isolated as previously described [2,3], with details specific to the work presented here provided in more detail. CuHARS were synthesized using copper sulfate and cystine at 37 °C, in 25 cm^2^ cell culture flasks, with multiple flasks being used to scale up the synthesis. Synthesized CuHARS mixtures were transferred to 15 mL or 50 mL centrifuge tubes, depending on the scale of the synthesis, and concentrated into a pellet by centrifugation. To ensure that any precipitates of copper not associated with CuHARS were destroyed, 3 µL of 0.1 M HCl was added per 10 mL of CuHARS solution. This mixture was then vortexed for 1 min to dissolve any precipitates. The tube was then centrifuged at 3000× *g*, the supernatant was decanted, and sterile water was added with vortexing to wash the pellet. The washing and centrifugation process was repeated a third time, and then as much of the supernatant was removed as possible without disturbing the CuHARS pellet, which was blue in color. For the determination of mass, this wet pellet of CuHARS was then transferred to a hot plate by adding CuHARS material to a glass microscope slide and dehydrating. Dried material was removed from the glass surface and weighed to calculate the yield and mass of the final product. Using these methods, and as described previously [1,2], the synthesis of CuHARS resulted in materials that ranged from 20 nm and larger in diameter, and hundreds of nm and more in length. 

AgCysNPs: Silver sulfate was reduced using the same amino acid, L-cystine, as previously described in the synthesis of CuHARS [2,3]. The entire synthesis of self-assembled AgCysNPs was carried out in 24-well cell suspension plates (VWR). After warming water and cystine solutions as previously described [2], a final concentration of 1 mM HCl was added to the mixture and warmed for an additional 30 min at 37 °C in a heating oven (VWR). Finally, the self-assembly reaction was initiated by addition of a final concentration of 0.4 mM silver sulfate, and the microplate returned to the heating oven. After 5 h of incubation time, the microplate was moved into refrigeration at 2–8 °C. The self-assembled nanoparticles were harvested after allowing at least 24 h of refrigeration. 

Quantification of Silver Nanoparticles: AgCysNPs prepared by self-assembly were centrifuged four times at 5000 relative centrifugal force (rcf) for 10 min in a 1 mL Eppendorf tube. After each run, the supernatant was discarded, and a 40-fold concentrated (by volume) slurry was obtained. This concentrated preparation was spotted on a glass slide, 30 µL at a time, and dried into a dry mass in a heating oven set at 37 °C until a measurable mass was obtained. Once completely dried, the particles were scraped off using a clean and sharp metal blade and carefully transferred into an anti-static weigh boat and weighed using a Mettler Toledo microbalance. The obtained AgCysNPs were then suspended in sterile deionized water to obtain a known concentration of 0.1 mg/mL and sonicated using a Branson 1800 sonicator under assisted monitoring and vortexing at time intervals for three hours to achieve a resuspension of brown colloidal solution of silver nanoclusters and nanoparticles. 

CuNPs were purchased from Sigma Aldrich (St. Louis, MO, USA) and handled as previously described [1]. Unoxidized copper nanoparticles (CuMNPs, 60–80 nm diameter) were purchased from SkySpring Nanomaterials, Inc. (Houston, TX, USA).

### 2.3. Characterization of Nanomaterials

Scanning Electron Microscopy (SEM, Hitachi, Japan): MOBs were imaged using a Hitachi FESEM by drying samples onto silicon wafers and then carrying out microscopy at the indicated magnification. 

Zeta Potential: Zeta potential of materials was taken as an average of at least three separate recordings of each sample. Each recording consisted of at least eight readings by using a Brookhaven Instruments Zeta Analyzer (Holtsville, NY, USA) by adding the sample to a cuvette filled with de-ionized water, and then immediately starting the recording session. 

### 2.4. Generation of Nanofilms In Vitro and Coffee Ring Effect/Image Analysis

Synthesized materials were spotted onto a 24-well cell culture (polystyrene) microplate by the addition of a 20 µL drop of material in water as indicated and allowed to dry at 37 °C in a heating oven (average time = 6 h), until all liquid had evaporated as verified by microscopy for evaluation of potential coffee ring effect. Control wells for cellular toxicity studies were spotted with 20 µL of water. 

### 2.5. Laser Imaging and Ablation of CuHARS

Multiphoton microscope imaging combined with the second harmonic generation (SHG) technique [15] was used to visualize unlabeled CuHARS. A dilute suspension of the material was dried on a microscope slide at room temperature. SHG was induced in the CuHARS using 890-nm excitation from a tunable Chameleon Vision-2, 80-MHz pulsed multiphoton laser (Coherent) and a VivoTM 2-photon microscopy workstation (Intelligent Imaging Innovations, Inc (3i), Denver, CO, USA) with an ELWD 40X, 0.6 NA microscope objective (Nikon, Japan) and GaAsP detectors controlled by Slidebook (3i) software. Laser power at the sample was adjusted using Pockels cell (Conoptics, Danbury, CT, USA) which was controlled through Slidebook. A 2-µs dwell time with pixel averaging (4/scan) was used for acquiring images. Emitted light was filtered with 458/64 nm BrightLine bandpass filter (Semrock, Inc., Rochester, NY, USA). For imaging CuHARS, laser power was set to 150 mW at the sample and it was briefly increased to 240 mW for ablation of the CuHARS.

### 2.6. Toxicity Testing against Cancer Cells: Quantification of Cell Death (or Survival)

Cancer cells at 40,000 cells/mL were plated onto 24-well cell culture microplates that had been previously spotted with nanofilm materials. After 3 days in vitro (3DIV) or as indicated, wells were terminated for MTT assay, which is an indicator for cellular viability and metabolism [14]. The MTT assay was carried out as we have previously described [14], with measured values normalized to control wells, which represented 100% viability.

In addition to this novel method of testing nanomaterial toxicity by forming nanofilms, materials were also tested for toxicity in a traditional manner, by adding to wells of glioma cells that had been previously adhered to the cell culture plate.

DAPI staining of cell nuclei after treating with nanomaterials was carried out as previously described [14,16,17,18]. Image ProPlus image analysis software was used to quantify the number of cell nuclei in each counted microscopic field. 

## 3. Results

A schematic overview of the synthesis of both CuHARS and AgCysNPs MOBS is shown in Figure 1 and Figure 2, respectively.

Self-assembled AgCysNPs formed stable colloids with solutions demonstrating clear color generation over time, with an as-synthesized product having a brown color in appearance (Figure 3A); after 90 min of settling time, the solution appeared unchanged (Figure 3B). In contrast, under the same conditions, CuHARS at 1 mg/mL after vortexing demonstrated a pale, blue color (Figure 3C), which clearly separated into bright blue MOBs at the bottom of the vial and a clear solution at top after 90 min of settling time (Figure 3D).

The size and shape of these two types of MOBs were also demonstrated using scanning electron microscopy (SEM). AgCysNPs were symmetrical, particle shaped, with average size of 40 nm, with a distribution from 20 to 80 nm, with larger nanoparticle aggregates also present (Figure 4A).

In contrast to nanoparticles, CuHARS had nanoscale diameters with a range of much longer lengths from hundreds of nanometers to microns (Figure 4B). The measured zeta potential of CuHARS recorded for comparative purposes with AgCysNPs, was (−28 mV to −33 mV), calculated from three different synthesized samples. In comparison, commercially available CuNPs used in this project were measured to have a zeta potential of +43 mV, as we previously described [1]. AgCysNPs synthesized here also had negative zeta potential values ranging from −16 mv to −23 mV. Both MOBs resulted in non-aggregating particulates, although AgCysNPs had a slight tendency to aggregate over time and could become highly compacted at the coffee ring perimeter during drying (see Figure 5). In contrast, CuHARS were remarkably non-agglomerating, possibly due to their repulsive negative zeta potential charge combined with their elongated shape.

The size and shape of Ag- and Cu-based MOBs generated here also resulted in very different biophysical behaviors when dried onto surfaces to form nanofilms. Both types of materials were diluted in water and then dried at 37 °C under sterile conditions. AgCysNPs demonstrated the well-known coffee ring effect (Figure 5A), while uncapped, copper oxide NPs agglomerated to a greater extent (data not shown).

In contrast to AgCysNPs, given the elongated, high-aspect ratio structure of CuHARS, these MOBs coated surfaces much more evenly, demonstrating very little coffee ring effect (Figure 6A,B). Furthermore, the elongated, long-range crystal-like structure of CuHARS was reflected in three optical properties of note not observed in the nanoparticles. First, these copper-based MOBs demonstrated polarization of white light as shown by use of traditional microscopy (Figure 6C–E).

As a second contrasting characteristic compared to AgCysNPs, when exposed to two-photon pulses using multi-photon microscopy techniques, the CuHARS demonstrated sufficient second harmonic generation (SHG) such that these MOBs could be illuminated (Figure 7A,B). Finally, sufficient multiphoton laser excitation was achieved such that CuHARS could be ablated by the absorbed laser light (Figure 7C).

To test the synthesized MOBs as potential toxic agents for anti-cancer treatments, we exposed Ag- and Cu-based MOBs to two cancer cell lines: CRL-2303s (glioma cells [1,19]) and SH-EP1, (a cell line subcloned from a neuroblastoma [20]). Non-capped CuNPs were used as positive controls for killing cells. For materials tested, we formed nanofilms of the material as described in methods, and then added cells for viability testing. The reactivity and toxicity responses were strikingly different for the two MOBs and for the CuNPs. The CuNPs formed nanofilms that quickly disappeared and killed cells under normal cell culture conditions (Figure 8A,B). In contrast, AgCysNPs formed compact, coffee ring structures that were retained for at least one day under cell culture conditions, and then began to degrade, but were potently toxic towards cancer cells (Figure 8A,B). AgCysNPs were significantly more toxic than CuNPs on a mass per volume basis (Figure 8A,B). The most stable of the three materials was the CuHARS, which slowly degraded over time in culture, and imparted moderate toxicity to cancer cells compared to AgCysNPs and CuNPs (Figure 8C,D).

In contrast to our nanofilms experiments (Figure 8), testing of nanomaterials and other substances for potential toxicity in vitro has been traditionally carried out by first plating the cells to be tested, and then adding the materials with potential toxic properties [1,12,13]. To compare the MOBs developed here also in traditional toxicity models, we added synthesized CuHARS and AgCysNPs to plated cancer cells and assessed toxicity (Figure 9).

We also compared quantified toxicity with free metal constituents used in the development of these MOBs, including unoxidized copper nanoparticles (Figure 10A) and silver sulfate (Figure 10B).

The assessment of in vitro toxicity of materials added to cells already plated was also quantified by staining for cell nuclei using DAPI, followed by using image analysis software (see Methods and Appendix A). As was the case for results using the MTT assay (Figure 9 and Figure 10), quantitative results using DAPI analysis indicated silver sulfate was the most potent material for causing cancer cell killing as compared to copper compounds (Appendix A).

## 4. Discussion

Previously we described the characterization and synthesis of self-assembled biohybrids composed of copper and the amino acid dimer cystine [1,2,3]. These copper-containing biohybrids, which we have named CuHARS, have very high-aspect ratio (linear) shape, presumably due to the cystine dimer which provides spacing between coordinated copper. From these studies we attempted to carry out self-assembly of silver nanoparticles using identical conditions as for the CuHARS, and achieved effective reduction and capping of silver sulfate to form silver nanoparticle colloidal solutions (AgCysNPs). In contrast to the CuHARS, synthesis of cystine with silver sulfate does not result in high-aspect ratio structures, presumably due to the +1 oxidation state of silver and interaction of silver with thiol groups, including those of cysteine [21].

To our knowledge, the work here is the first report of direct synthesis of silver nanoparticles using only cystine as the capping/reducing agent. Zhang et al. recently utilized L-cysteine to assist with the synthesis of silver nanoparticles, but in all cases, it was in the presence of sodium citrate, with in some cases the cysteine being completely absent [22]. Thus here, we show the direct formation of AgCysNPs with the amino acid dimer alone, and under self-assembling conditions at body temperature (37 °C). Since both copper and silver could be directly used to form metal–organic biohybrids (MOBs), we compared the two composites for material-cell interaction purposes. Both silver and copper have been shown to have anti-cancer applications [23,24,25], so we therefore compared the newly synthesized MOBs for toxicity towards cancer cells.

The synthesized AgCysNPs were extremely potent towards killing of both glioma and SH-EP 1 cells (Figure 8A,B, respectively). When compared directly with CuNPs on a mass/volume basis, AgCysNPs were at least 10-fold more potent than the copper nanoparticles for SH-EP 1 cells (Figure 8B, comparing viability of 2000 ng/mL CuNPs vs. 200 ng/mL AgCysNPs), and at least 20-fold more potent for glioma cells (Figure 8A, comparing 2000 ng/mL CuNPs vs. 100 ng/mL AgCysNPs). As expected, silver MOBs were more reactive and broke down faster under cell conditions compared to CuHARS, due to the nanoparticle nature of the silver biocomposites (as shown by SEM and coffee ring behaviors). In contrast, copper MOBs as synthesized here were of a high aspect ratio structure and took a number of days to fully degrade under the cell culture conditions tested here and as recently described [3]. This was reflected as well in the toxicity of CuHARS against both SH-EP 1 cells (Figure 8D) and glioma cells (Figure 8C). When comparing the toxicity towards SH-EP 1 cells, CuHARS at 5000 ng/mL (Figure 8D) was 25-fold less toxic than AgCysNPs (Figure 8B, 200 ng/mL). When comparing the toxicity towards glioma cells, CuHARS at 10,000 ng/mL (Figure 8C) was 100-fold less toxic than AgCysNPs (Figure 8A, 100 ng/mL). 

Thus, the shape (nanoparticle vs. CuHARS), and composition (silver vs. copper), of different MOBs may provide utility in future studies for delivering fast-acting and delayed-acting features when interacting with cells. Furthermore, metal salts, while demonstrating toxicity in the case of silver sulfate (Figure 10B), would also be expected to be fast acting, since they are completely soluble. This characteristic may not allow for delayed action, since all of the material starts out in solution. In fact, in the studies shown here, silver sulfate was less toxic to glioma cells on a mass/volume basis (Figure 10B) than were the synthesized AgCysNPs (Figure 9A). For the copper materials, as expected, both oxidized CuNPs and unoxidized CuMNPs were more toxic than the amino acid containing CuHARS (Figure 9B and Figure 10A). Additionally, when copper sulfate salt was evaluated at 100 ng/mL for toxicity towards glioma cells, only a 19.5% reduction in viability was measured (data not shown) compared to 52% reduction in viability measured for silver sulfate at the same concentration (Figure 9A). This result may be due in part to the endogenous copper chelating capabilities of albumin and ceruloplasmin found under cell culture conditions [3,9,10].

In this first report of our newly synthesized AgCysNPs, it appears that compared to the numerous other types of silver nanoparticles synthesized and tested for anticancer activity, that our nanoparticles are extremely potent, with an estimated IC_50_ of 500 ng/mL against SH-EP 1 cells (Figure 8B) and an even more potent action with an estimated IC_50_ of 200 ng/mL against glioma cells (Figure 8A). In comparison from the literature, He et al. [26] reported an IC_50_ of slightly less than 10,000 ng/mL against PC-3 cells, and Mittal et al. reported an IC_50_ of 100,000 ng/mL for AgNPs against a Dalton lymphoma cell line [27]. While differential responses of silver nanoparticles may be expected due to different synthesis methods and sensitivity of different cell types, AsaRani et al. reported the inhibition of cell viability in the 25–50 µg/mL range for silver nanoparticles in a direct comparison of normal human fibroblasts and glioblastoma cells [28]; normal fibroblasts were shown to be more resistant to the nanoparticles than the cancer cells in these studies. 

The silver and copper MOBs synthesized here were both fully biodegradable [3], which provides benefits when developing materials for cell interactions. Due to the linear, regular array and crystalline-like nature of the CuHARS, we found that individual copper MOBs could be identified and imaged via SHG with two-photon microscopy, and indeed, at higher laser power, the individual structures could be ablated by the laser beam absorption (Figure 7). This is consistent with previous studies describing SHG in crystals of tris(thiourea)copper(1) perchlorate [29], but has not been previously described for the CuHARS. Thus, metal content and biocomposite 2D- and 3D-structures may assist in assembling and degrading MOBs for applications to cells. Two-channel multiphoton microscopy (MPM) images comprised of SHG images of CuHARS and standard MPM images of fluorescently labelled cells will facilitate localization of CuHARS within cells and tissue for in vitro and in vivo assays, respectively. For the in vivo case, this could enable non-invasive and minimally invasive studies of tumor penetration and reduction of tumor volume over time in live animal models. Intentional degradation of CuHARS by focused multiphoton laser beam at slightly higher power may be an effective means of providing targeted, bolus release of Cu for accelerating cancer death.

Above and beyond the application of metals-based nanoparticles (including silver and copper) in nanomedicine, which are indeed numerous [30,31,32], it is envisioned that metal–organic biohybrids (MOBs) that take on different sizes and shapes will provide an array of capabilities for cell interactions, both for external body (surface) and internal body exposures. The well-known coffee ring effect [11], for example, may be used to demonstrate how synthesized MOBs of different sizes and shapes can be used to form different types of nanofilms onto surfaces. As shown here, using two completely biodegradable MOBs, the resulting breakdown and/or toxicity of the materials formed into such nanofilms may be used to study different kinetics towards, for example, killing cancer cells. None of the MOBs used here were functionalized for specific targeting or for synergistic killing using known cancer drugs. Thus, the toxicity quantified here is intrinsic to the biohybrid materials themselves as synthesized. Functionalization of nanomaterials is a fertile area of nanoparticle-assisted cancer research [33,34,35,36] and may thus be utilized in future studies to alter silver and copper MOBs self-assembled using cystine.

## Figures and Tables

**Figure 1 nanomaterials-09-01282-f001:**
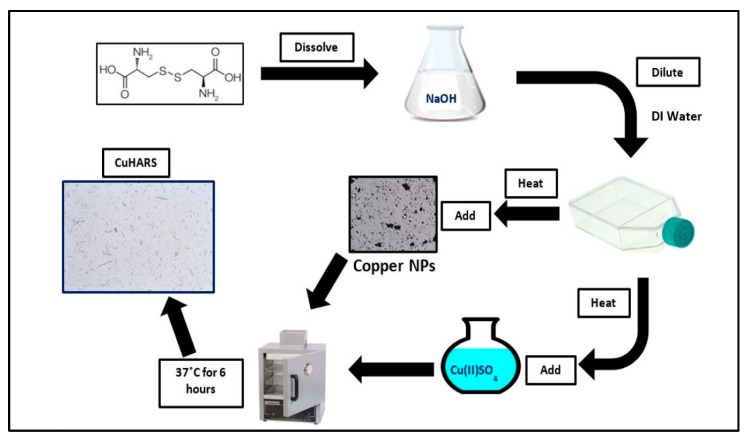
Schematic overview of synthesis of copper-containing high-aspect ratio structures (CuHARS) metal–organic biohybrids (MOBs). In the work presented here, synthesis was carried out only using copper sulfate (Cu(II)SO_4_).

**Figure 2 nanomaterials-09-01282-f002:**
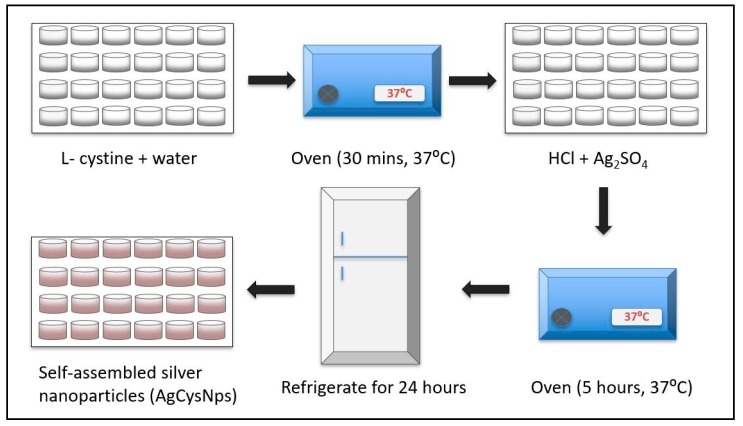
Schematic overview of synthesis of cystine-capped silver nanoparticles (AgCysNPs) MOBS. AgCysNPs were synthesized in microwell plates as indicated, and after completing the self-assembly step in refrigeration, were harvested, and dried for mass determination and use with in vitro testing as described in the methods.

**Figure 3 nanomaterials-09-01282-f003:**
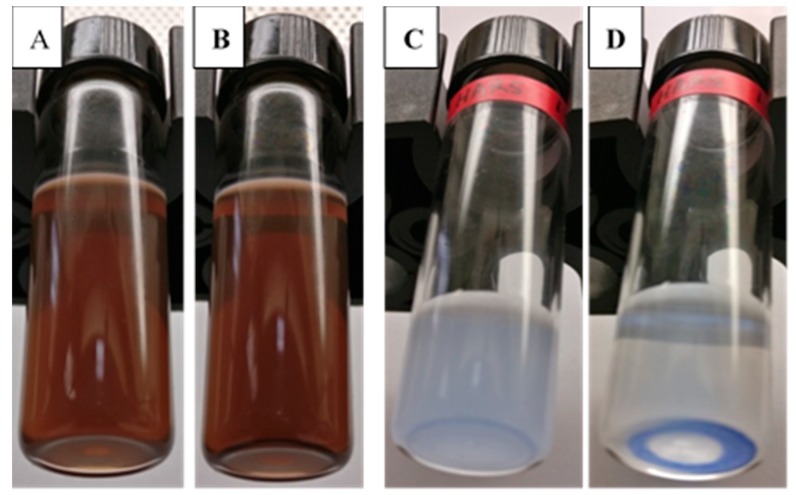
Properties of MOBs in solution: AgCysNPs (**A**,**B**) as synthesized or CuHARS (**C**,**D**) at 1 mg/mL were vortexed, followed by 90 min of settling time at room temperature (**B**,**D**).

**Figure 4 nanomaterials-09-01282-f004:**
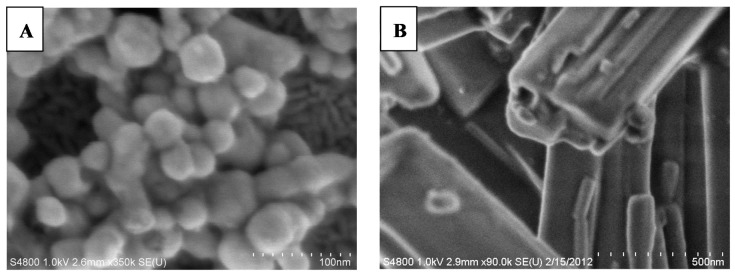
Scanning electron micrographs of synthesized MOBs: (**A**) AgCysNPs, scale bar indicates 100 nm with 10 nm gradations; (**B**) CuHARS, scale bar indicates 500 nm with 50 nm gradations.

**Figure 5 nanomaterials-09-01282-f005:**
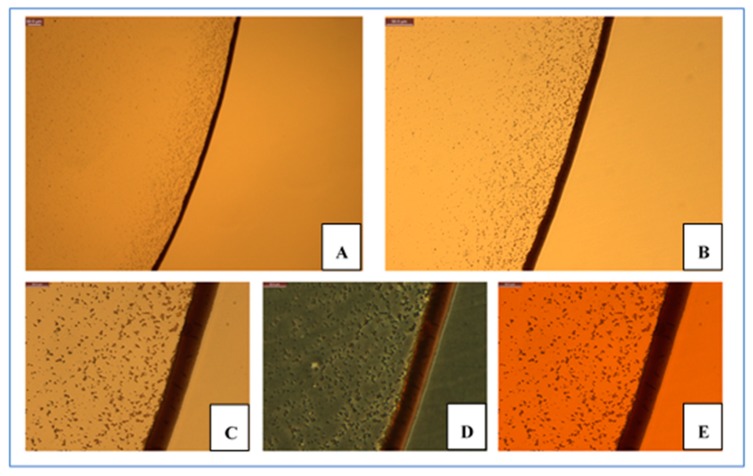
Microscopy characterization of AgCysNP dried films. Nanofilms were prepared as described in methods. Nanoparticles formed a distinct coffee-ring effect as shown under white-light microscopy using 10× and 20× objectives (**A**,**B**, respectively). Zoomed portions of the dried films were further characterized using bright-field, polarization, and phase microscopy (panels **C**–**E**, respectively). Scale bars in upper left of each image indicate 50 microns for panels **A**,**B**; and 20 microns for panels **C**–**E**.

**Figure 6 nanomaterials-09-01282-f006:**
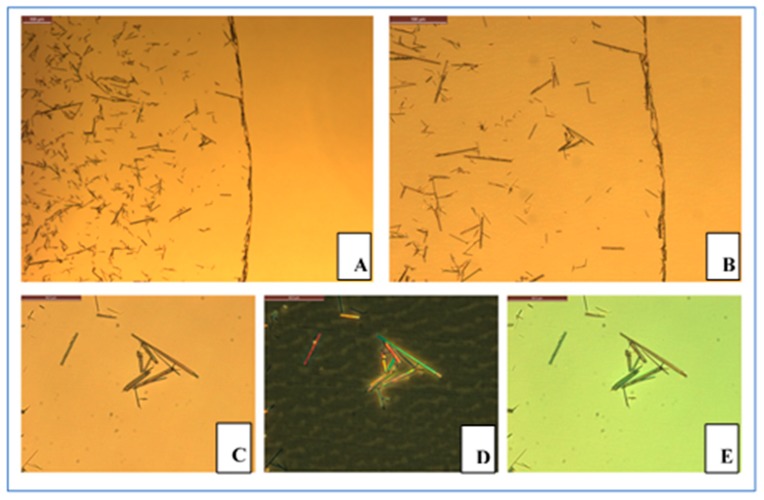
Microscopy characterization of CuHARS dried films. CuHARS were applied to 48-well culture plates as for AgCysNPs, and then imaged using brightfield microscopy with 10× and 20× objectives (**A**,**B**, respectively). Additionally, zoomed areas were imaged using bright-field, polarization, and phase microscopy (panels **C**–**E**, respectively). Scale bars in upper left of each image indicate 100 microns for panels **A**,**B**, and 50 microns for panels **C**–**E**.

**Figure 7 nanomaterials-09-01282-f007:**
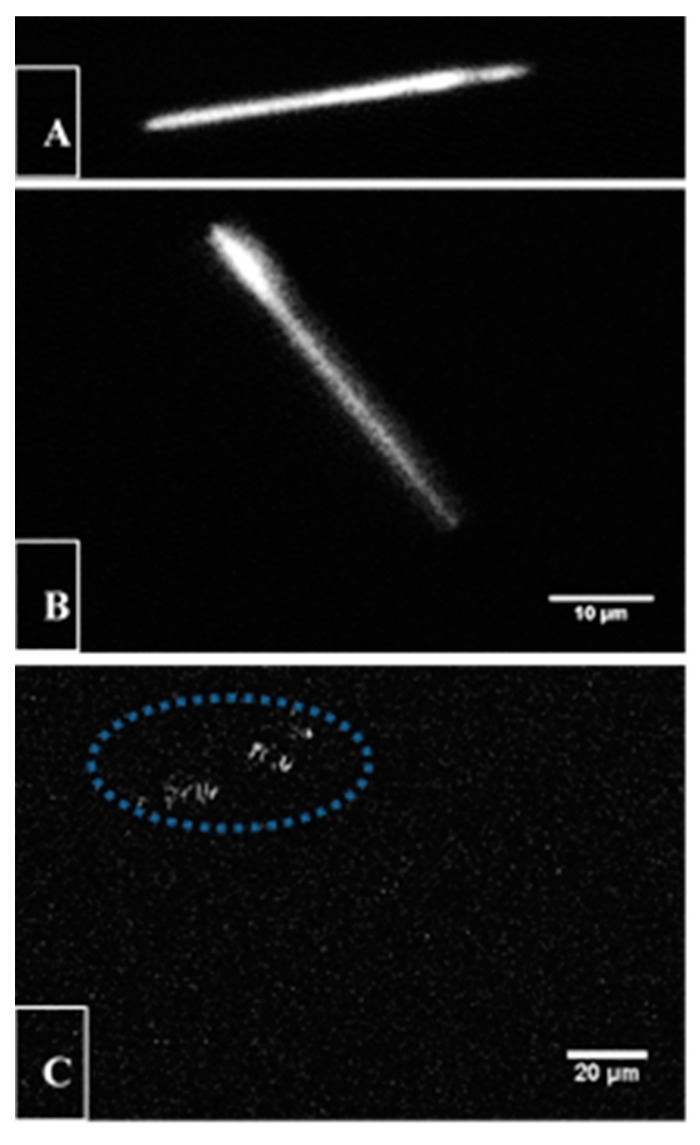
Optical properties of CuHARS. SHG induced in the CuHARS using 890-nm, scanned, pulsed laser light from a multiphoton laser. (**A**,**B**): Two individual CuHARS lying on a glass microscope slide with a No. 1.5 glass coverslip are shown. (**C**) Rapid degradation of single CuHARS using higher laser power. Only small, particulate-like material remains after photo destruction of the CuHARS (remaining particulates are inscribed by dotted ellipse). Scale bars are 10 microns for panels **A**,**B**, and 20 microns for panel **C**.

**Figure 8 nanomaterials-09-01282-f008:**
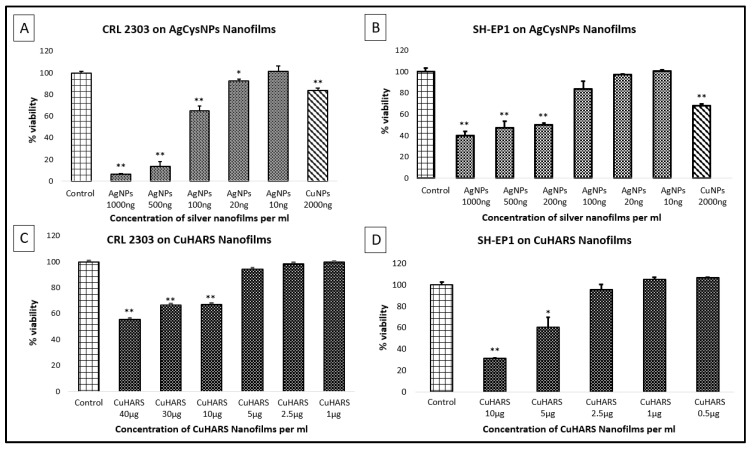
Toxicity testing of nanomaterials as films. CRL 2303 glioma cells at 40,000 per well were treated with AgCysNPs or CuNPs (Panel **A**), at the indicated concentrations, or CuHARS (Panel **C**), and tested for viability. SH-EP1 cells at 40,000 per well were grown in vitro for three days on nanofilms of the indicated materials, either AgCysNPs (Panel **B**), or CuHARS (Panel **D**). Upon termination, an MTT assay was carried out to test for cellular metabolic activity. Data for each condition are the average of *n* = 3 wells with standard error of the mean indicated. Data represent multiple platings of each cell type tested, with * denoting *p* < 0.05 and ** denoting *p* < 0.01 compared to control conditions. *Y*-axis = Viability normalized to % of controls (column 1 values). For evaluation of AgCysNPs (Panels **A**,**B**), the right-most column = addition of CuNPs for comparative purposes at 2000 ng/mL. Controls were untreated cells (Panels **A**–**D**).

**Figure 9 nanomaterials-09-01282-f009:**
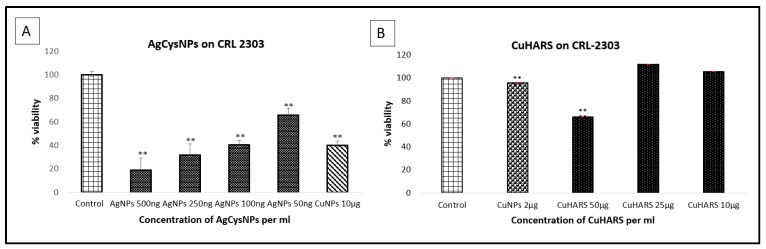
Toxicity testing of nanomaterials added on incubated cells. CRL 2303 glioma cells at 40,000 per well were treated with AgCysNPs or CuNPs (Panel **A**), at the indicated concentrations, or CuHARS (Panel **B**), and tested for viability. CRL 2303 cells at 40,000 per well were grown in vitro for two days and then treated with either AgCysNPs (Panel **A**), or CuHARS (Panel **B**). Upon termination after 24 h post treatment, an MTT assay was carried out to test for cellular metabolic activity. Data for each condition are the average of *n* = 2 wells with standard deviation indicated. Data represent multiple platings of each cell type tested, with ** denoting *p* < 0.01 compared to control conditions. *Y*-axis = Viability normalized to % of controls (column 1 values). For evaluation of AgCysNPs (Panel **A**), the right-most column = addition of CuNPs for comparative purposes at 10,000 ng/mL. Controls were the untreated cells in both panels.

**Figure 10 nanomaterials-09-01282-f010:**
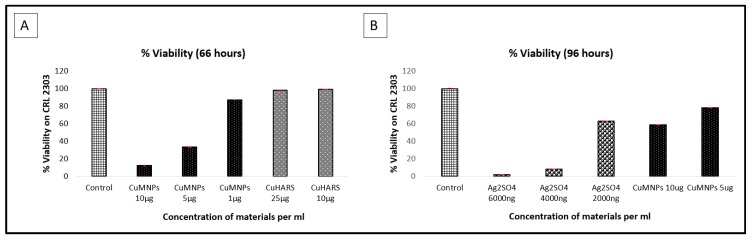
Toxicity testing of free metals added on incubated cells. CRL 2303 glioma cells at 40,000 per well were treated with unoxidized copper nanoparticles (CuMNPs) and CuHARS (Panel **A**), and at the indicated concentrations of Ag_2_SO_4_ and CuMNPs (Panel **B**) and tested for viability. CRL 2303 cells at 40,000 per well were grown in vitro for two days and three days respectively as shown in panel A and B and then treated with either CuMNPs and CuHARS (Panel **A**), or Ag_2_SO_4_ (Panel **B**). Upon termination after 18 h (Panel **A**) and 24 h (Panel **B**) post treatment, respectively, an MTT assay was carried out to test for cellular metabolic activity. Data for each condition are the average of *n* = 2 wells, representing multiple plates from multiple platings of cells. *Y*-axis = viability normalized to % of controls (column 1 values). Controls were the untreated cells in both panels.

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
