# Peer review of "Self-Assembled Metal–Organic Biohybrids (MOBs) Using Copper and Silver for Cell Studies"

_nanomaterials, 2019, doi:10.3390/nano9091282_

Round 1

Reviewer 1 Report

The Karekar et al., fabricated Self-assembled metal-organic biohybrids (MOBs) for cell studies. The study looks compatible with the journal and might be accepted after addressing the following comments;

English grammar need to be improved The abstract should include some data values obtained from experiments, not just discussions. Introduction section needs to be improved with related literature. Detailed synthesis procedure will be helpful for the reader’s not just references. Authors need to give very suitable graphical abstract. Schematic presentation of synthesis will helpful for the readers to a quick understanding of the material. Sen tense ‘The silver and copper MOBs synthesized here were both fully biodegradable, which provides 236 benefits when developing materials for cell interactions’ needs suitable reference. The obtained results can be discussed in further details. The comparison of the potency of material in the following para ‘In this first report of our newly synthesized AgCysNPs, ……………….234 Dalton lymphoma cell line [18]’ is not correct as authors compared AgNPs potency with different cell lines.

Author Response

We thank the reviewer for the helpful comments. The abstract has now been updated to include some data values as suggested. the introduction has now been improved to include also related literature and this includes additional references.   detailed synthesis procedure has now been included as suggested. A graphical abstract has been created and is ready to submit as part of the overall manuscript. A new figure has been created (figure 1), which gives a schematic presentation of synthesis as suggested. the sentence previously appearing on line 236 about biodegradable MOBs has been given a suitable reference. The obtained results have now been discussed in further detail, with added text in the discussion section. We agree with the reviewer that comparison of potency only to lymphoma cells would not be correct, since we used glioma cells. We have therefore now added some discussion and a reference to work done also in glioblastoma.  

We have clearly marked where major changes have been made by color highlighting.  

Reviewer 2 Report

The usage of fabricated nanomaterial is not clear. Even though the process of synthesizing the nanomaterial is novel, the anti-cancer effects of the material should be evaluated properly by seeding cells in a tissue culture plate, followed by incubation of the cells in presence of the synthesized nanoparticles. Moreover, it should be demonstrated whether the cytotoxic effects are due to the formed nanoparticles or free metal. 

Author Response

1.  we thank the reviewer for the helpful comments.

2. We have taken the reviewer's advice and now added new data from experiments where the anti-cancer effects were quantified by first seeding the cells followed by incubation of the cells in the presence of the synthesized nanoparticles.  We feel this included comparison now strengthens the manuscript.  

3.  We have also carried out new experiments as suggested to also quantify the cytotoxic effects of free metal constituents used to make our novel nanomaterials:  thus, we have provided data from new experiments using silver sulfate and copper sulfate salts, as well as unoxidized copper nanoparticles.  We agree with the suggestions of the reviewer and we believe these new provided data help strengthen the impact of the manuscript.

we have clearly marked where major changes have been made by color highlighting.  

Round 2

Reviewer 2 Report

Manuscript can be accepted since the authors could address the major issues raised.